# Tissue Culture via Protocorm-like Bodies in an Orchids Hybrids *Paphiopedilum* SCBG Huihuang90

**DOI:** 10.3390/plants13020197

**Published:** 2024-01-11

**Authors:** Beiyi Guo, Hong Chen, Yuying Yin, Wei Wang, Songjun Zeng

**Affiliations:** 1 Guangzhou Collaborative Innovation Center on Science-Tech of Ecology and Landscape, Guangzhou Landscape Plant Germplasm Resource Nursery, Guangzhou Institute of Forestry and Landscape Architecture, Guangzhou 510540, China; gzifla_gby@gz.gov.cn (B.G.); gzifla_ch@gd.gov.cn (H.C.); 2Key Laboratory of South China Agricultural Plant Molecular Analysis and Gene Improvement, South China Botanical Garden, Chinese Academy of Sciences, Guangzhou 510650, China; yinyuying18@mails.ucas.edu.cn

**Keywords:** protocorm-like body, in vitro propagation, *Paphiopedilum*

## Abstract

This study successfully established an efficient in vitro propagation system for *Paphiopedilum* SCBG Huihuang90 via protocorm-like body (PLB) formation from seed-derived calluses, PLB proliferation and differentiation, root induction and greenhouse acclimatization. Furthermore, 1/2 Murashige and Skoog (MS) + 0.025 mg/L 2,4-Dichlorophenoxyacetic acid (2,4-D) was suitable for the proliferation of PLBs, and 1/2MS + 10% coconut water (CW, *v*/*v*) + 0.5 g/L activated carbon (AC) was suitable for PLB differentiation. PLBs at different developmental stages required different kinds of sugars. This study provided a reference for research on the propagation techniques of other *Paphiopedilum*.

## 1. Introduction

*Paphiopedilum* Pfitzer (Orchidaceae) is usually known as the slipper orchid, which is named for its unique pouch-shaped lip similar to ladies’ slippers. Because of its special flower shape and high commercial value, many *Paphiopedilum* species have been over-exploited [1]. All wild *Paphiopedilum* species are listed in the Convention on International Trade in Endangered Species of Wild Fauna and Flora (CITES) Appendix I. So far, most of the commercial production of *Paphiopedilum* has been through the asymbiotic germination of seeds, and few varieties have achieved asexual cloning [1,2,3]. This is detrimental to obtaining commercialized seedlings with consistent traits.

Tissue cultures have the advantages of high breeding efficiency, no seasonal limitation, and the maintenance of excellent parental traits. It has been widely used for the large-scale breeding of ornamentals. Therefore, the establishment of an efficient breeding system is of great significance for the protection of *Paphiopedilum* and for meeting market demand.

*Paphiopedilum* is considered to be difficult to propagate through in vitro tissue culture because it is difficult to induce direct organogenesis or callus from mature explants such as leaves, stem segments, and roots [1,4,5,6]. In addition, the callus of *Paphiopedilum* has a low proliferation or regeneration capacity [3,7]. 

PLBs, which are derived from somatic cells during in vitro tissue cultures, are similar to orchid protocorm in morphology and biological characteristics [8,9]. PLBs are ideal explants for micropropagation [10] because of their ability to differentiate into apical meristem and complete plantlets. In orchids, the formation of PLB is regulated by various factors, and plant growth regulators (PGRs) are one of the most important factors [8]. Thidiazuron (TDZ), 2,4-D, or their combination were considered to be the most suitable PRGs for callus and PLB induction [8,11]. The induction of PLB formation by 2,4-D or TDZ has been confirmed to be successful in some orchids such as *Cymbidium* [12], *Phalaenopsis* [13], and *Dendrobium* [14]. However, the formation of *Paphiopedilum* PLBs is difficult. According to the reports, so far, only five species of *Paphiopedilum* have successfully induced PLBs from callus [3,7,15,16,17,18] and one from protocorm [5]. The regeneration of PLBs is a complex morphogenetic process that involves many biological events, including changes in the content of various metabolites. Sugars play an important role in plant life activities and may affect the formation of PLBs. No studies have been conducted to investigate the changes in the content of sugars during the development of PLBs in *Paphiopedilum*.

The goals of this study are to deepen the understanding of the formation and development conditions of *Paphiopedilum* PLBs and to establish an effective propagation system via PLBs in *P.* SCBG Huihuang90. 

## 2. Results

### 2.1. Callus Induction and Proliferation

About 53% of seeds germinated into protocorms in the seeding medium. After 60–90 days of continued culture on the same medium, some of the protocorms gradually expanded to form calluses, which were easily differentiated. On each of the test media, these calluses could not be maintained or proliferate. After 15 days of culture, external calluses began to differentiate into PLBs. The differentiation of different parts of a callus was asynchronous, and there were undifferentiated cells within the calluses, resulting in a mixed mass of calluses and PLBs. In each test medium, with the extension of culture time, by 30 d of culture, PLBs covered the surface of the surviving calluses and became PLB masses. PLB masses could be used for subsequent proliferation and differentiation cultures.

TDZ and higher concentrations of 2,4-D (1.0–2.0 mg/L) treatments increase the mortality (Table 1). When the concentration of 2,4-D or TDZ reached 1.5 mg/L, the surviving explants grew poorly, with white color and slow growth. Moreover, different concentrations of 2,4-D or TDZ failed to maintain and proliferate calluses the of *P.* SCBG Huihuang90. The calluses differentiated and grew well on the medium 1/2MS + 0.5 mg/L 2,4-D as well as on 1/2MS.

### 2.2. Effect of 2,4-D and TDZ on PLB Proliferation

Since the individual PLBs were too small and easily died after being cut off, a mass of PLBs was an ideal explant for proliferation. Concentrations of 2,4-D or TDZ had an effect on the proliferation efficiency of the PLB mass. On the one hand, 2,4-D significantly promoted the proliferation of PLBs. The highest proliferation coefficient and lowest mortality was obtained in the 1/2MS medium supplemented with 0.025 mg/L 2,4-D, and the proliferative coefficient was about 2.17 times higher than that of CK. As the concentration of 2,4-D increased, the proliferation rate decreased (Table 2). Visibly, PLBs of *P.* SCBG Huihuang90 were sensitive to 2,4-D. On the other hand, the higher concentration of TDZ (0.25–0.5 mg/L) promoted the proliferation of PLBs but resulted in significantly higher mortality rates. The combination of 0.05 mg/L 2,4-D and 0.5 mg/L TDZ inhibited the proliferation of the PLB mass and caused significantly increased mortality rates (Table 2).

The morphology of the PLB mass under the TDZ and 2,4-D treatment was different. The PLB mass was barely differentiated under 0.01–0.05 mg/L 2,4-D treatment and grew well, showing a yellowish green to light green color (Figure 1A). The individual globular PLBs were small in size and were closely arranged on the surface of the PLB mass (Figure 1B). The PLB mass was light green to dark green and most explants turned brown with the treatment of TDZ (Figure 1C). The individual globular PLBs were larger, spherical, or an elongated ellipsoidal shape (Figure 1D).

In summary, the 1/2MS medium supplemented with 0.025 mg/L 2,4-D was the optimum proliferation medium for the *P.* SCBG Huihuang90 PLB mass.

### 2.3. Effect of Different Media on the Differentiation of PLBs 

In CK, the differentiation rate of the PLB mass was low and the browning rate was high. Browning was found to occur generally at the incision of the PLB mass, and the area of the browning site did not continue to expand after 30 d of incubation. The no-browning part grew well and was able to differentiate normally. Browning causes the poor growth status of the PLB mass.

Low concentrations of BA (0.25 mg/L) or kinetin (KT) did not improve the differentiation rate of the PLB mass and exacerbated their browning. Furthermore, 0.5 mg/L BA significantly increased the differentiation rate and browning rate of the PLB mass (Table 3). Additionally, 0.5 g/L AC could significantly reduce the browning rate but did not contribute to the differentiation rate. The combination of 10% CW (*v*/*v*) and 0.5 g/L AC showed the highest differentiation rate and the lowest browning rate (Table 3).

The degree of differentiation of PLB masses varied after 60 days of culture. One way this occurred was when the whole mass of PLBs fully differentiated into cluster shoots (Figure 2A); the other was when part of the PLBs differentiated into cluster shoots, while the other part remained in an undifferentiated state (Figure 2B). To further analyze the efficiency of differentiation, the number of shoots formed by one PLB mass was compared. As shown in Table 3, at the time of 60 d of culture, the highest number of shoots formed were achieved under 10% CW (*v*/*v*) + 0.5 g/L AC treatment. In summary, it can be concluded that the most suitable medium for PLB differentiation is 1/2MS + 10% CW (*v*/*v*) + 0.5 g/L AC.

### 2.4. Effect of 6-BA, TDZ, KT and 2,4-D on Cross-Cut Leaves of P. SCBG Huihuang90 in In Vitro Culture

Neither organogenesis nor calluses could be successfully induced in the cross-cut leaves of *P.* SCBG Huihuang90 on CK as well as on the six test media. After 5 d of culture, browning began to occur on the edges of the cross-cut leaves. After 30 d of culture, the mortality rate of the cross-cut leaves was higher than 95% in all treatments. After treatment with TDZ, 2,4-D, 1.0 mg/L BA, and 1.0 mg/L KT, the cross-cut leaves turned brown and the mortality rate was 100% (Figure 3). Although in the other treatments there were still green cross-cut leaves at 30 d, no organogenesis or callus formation was seen (Figure 4A,B). After 60 d, all cross-cut leaves in each treatment finally turned brown and died.

### 2.5. Intact Leaves of Seedlings in In Vitro Culture

Neither organogenesis nor calluses could be successfully induced in the intact leaves of *P.* SCBG Huihuang90 on CK as well as on the six media. After 30 d of culture, all intact leaves turned brown and died (Figure 4C,D).

**Figure 4 plants-13-00197-f004:**
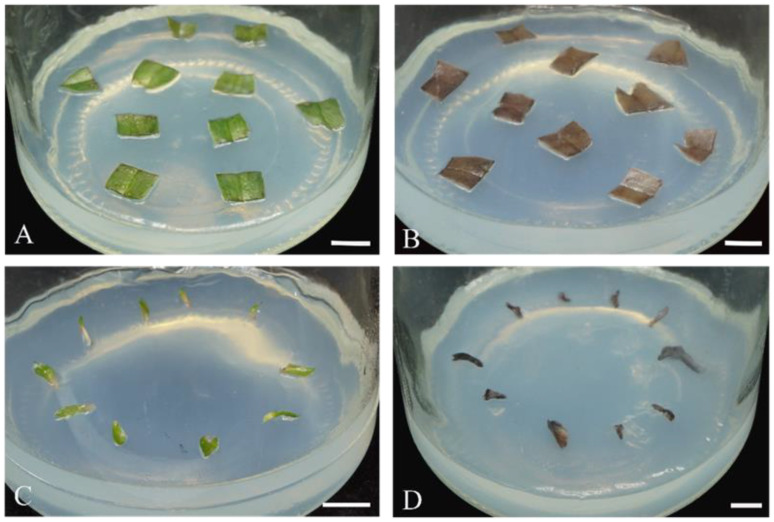
Tissue culture of plantlet leaves in in vitro of *P.* SCBG Huihuang90. (**A**) The cross-cut leaves are inoculated in medium 1/2MS + 0.5 mg/L 2,4-D at 0 d. (**B**) The cross-cut leaves are inoculated in medium 1/2MS + 0.5 mg/L 2,4-D at 30 d. (**C**) Leaves of the young seedlings are inoculated in medium 1/2MS+0.5 mg/L 2,4-D at 0 d. (**D**) Leaves of the young seedlings are inoculated in medium 1/2MS+0.5 mg/L 2,4-D at 30 d. (**A**–**D**: Scale bar = 1 cm).

### 2.6. Root Induction and Seedlings Acclimatization

Concentrations of 1-Naphthaleneacetic acid (NAA) affected the rooting stage of explants evidently. The 0.5 mg/L NAA treatment resulted in the longest average root length (2.06 cm) and the highest average number of roots (3.22). Moreover, higher concentrations of NAA (1.0–2.0 mg/L) treatment did not further improve the rooting rate and root numbers but rather inhibited their root elongation (Table 4). In addition, the rooting rate of all treatments was 100% at 120 days of rooting culture, and the seedlings grew well.

In conclusion, the optimum medium for *P.* SCBG Huihuang90 rooting induction is 1/2MS + 0.5 mg/L NAA + 0.5 g/L AC. Seedlings that had rooted and were in good growth conditions (Figure 5A) were suitable for transplantion. After 90 days, 96% of plantlets survived, and new leaves grew (Figure 5B).

### 2.7. Change in Carbohydrate Content in the Regeneration of PLB Masses 

The regeneration process of PLBs of *P.* SCBG Huihuang90 was divided into four stages with reference to previous morphological studies (Guo et al., 2021). As shown in Table 5, the overall trend of the soluble sugar content during the regeneration of PLBs of *P.* SCBG Huihuang90 was that it rose and then declined. The soluble sugar content was lower in the callus stage and increased significantly by about 2.29 times with the formation of PLBs. With the differentiation of PLBs, the soluble sugar content gradually decreased, and the soluble sugar content of the mixture of shoots and PLBs stage was significantly lower than that of the PLB mass stage and significantly higher than that of the seedling stage.

During the regeneration process of PLBs of *P.* SCBG Huihuang90, the overall trend of the starch content was that it increased first and then decreased (Table 5). As the calluses differentiated into the PLB mass, the starch content increased significantly and then remained stable. There was no significant difference in the starch content between the PLB mass stage and the mixture stage. The starch content decreased significantly from the mixture stage to the seedling stage.

## 3. Materials and Methods

### 3.1. Plant Material

The *P.* SCBG Huihuang90 was a hybrid *Paphiopedilum* bred by our research group and planted in the South China Botanical Garden, Guangzhou, China. Its seed parent was *P.* SCBG Prince, registered with The International Orchid Register (http://apps.rhs.org.uk/horticulturaldatabase/orchidregister/orchiddetails.asp?ID=972703) on 21 February 2014, and its pollen parent was named *P.* SCBG Miracle, registered with The International Orchid Register (http://apps.rhs.org.uk/horticulturaldatabase/orchidregister/orchiddetails.asp?ID=963533) on 8 March 2013. After 190–220 days of artificial pollination, mature seed capsules of hybrid materials were collected. The seeds were sown aseptically in a seed germination medium with reference to the method of Guo [6]. Referring to Zeng [19], the seed germination medium was Hyponex N026 medium supplemented with 0.5 g/L activated carbon and 1 mg/L NAA. The seedlings formed via seed germination were culled, and seed-derived calluses were used for subsequent experiments.

### 3.2. Media Preparation and Culture Conditions of In Vitro Culture

1/2 MS medium, containing half-strength macro-elements of MS salts [20], was used as the basal medium in all experiment. To each medium, 30 g/L sucrose, 5.8 g/L agarose and corresponding PGRs according to the experiment were added. The pH of all media was adjusted to 5.8–5.9. Each glass flask with a diameter of 10.5 cm contained fluid nutrient media and were sterilized with autoclave at 121 °C for 20 min.

All explants were incubated in a culture room at 25 ± 1 °C and a 12/12 h (light/dark) photoperiod with 2500–3000 lux under cool white fluorescent tubes.

### 3.3. Effect of 2,4-D and TDZ on Callus Culture

Seed-derived calluses were used for the explants and cultured on the basal medium supplemented with different concentrations of 2,4-D (0.5, 1.0, 1.5, 2.0 mg /L) or TDZ (0.5, 1.0, 1.5, 2.0 mg/L). PGR-free 1/2 MS medium served as the control (CK). Each treatment had three independent replicates, and one replicate consisted of 45 explants.

After 30 days, the differentiation rate (differentiation rate = 100 × number of differentiated calluses/number of inoculated calluses) and mortality rate (mortality = 100 × number of dead calluses/number of inoculated calluses) were calculated.

### 3.4. Effect of 2,4-D and TDZ on Proliferation of PLBs 

The mass of PLBs was cut into 50–60 mg weight and then cultured on the basal medium supplemented with different concentrations of 2,4-D (0.01, 0.025, 0.05 mg /L) or TDZ (0.025, 0.05, 0.1, 0.25, 0.5 mg /L), separately, and on the basal medium supplemented with both 0.05 mg/L 2,4-D and 0.5 mg/L TDZ. PGR-free half-strength MS medium served as the control (CK). Each treatment had three independent replicates, and one replicate consisted of 60 explants.

After 60 days, the proliferation rate [proliferation rate = weight of PLBs (mg)/weight of original explant (mg)] and mortality rate (mortality rate =100 × number of dead PLBs/number of inoculated PLBs) were calculated.

### 3.5. Effect of Additive on Differentiation of PLBs 

PLB masses with a weight of 40–50 mg were selected as explants. PLB differentiation medium were as follows: (1) 1/2MS + BA (0.25, 0.5 mg/L), (2) 1/2MS + KT (0.25, 0.5 mg/L), (3) 1/2MS + 0.5 g/L AC, and (4) 1/2MS + 10% CW (*v*/*v*) + 0.5 g/L AC. PGR-free half-strength MS medium served as the control (CK). Each treatment consisted of three independent replicates, and one replicate consisted of 60 explants.

After 60 days, the differentiation rate (differentiation rate = 100 × number of differentiated explants/numbers of inoculated explants), browning rate (browning rate = 100 × number of browning explants/number of inoculated explants) and the number of shoots per PLB mass were calculated.

### 3.6. Effect of NAA on Root Induction

When the leaves of seedlings reached 2 cm, the cluster shoots were divided into individual plants and used as explants for the rooting induction. To evaluate the effect of NAA on rooting induction, seedlings were cultured on basal medium supplemented with 0.5 g/L AC and different concentration of NAA (0, 0.1, 0.5, 1.0, 2.0 mg/L). Each treatment consisted three independent replicates, and one replicate consisted of 100 explants. After 60 days, the rooting rate (rooting rate = 100 × number of rooting explants/number of inoculated explants) and rooting numbers were calculated.

### 3.7. Cross-Cut and Intact Leaves of Seedlings in In Vitro Culture

Leaves in two different states were used as explants. In the first one, seedlings with a leaf length up to 3 cm were selected, and their leaves were cut into small segments of about 1 cm along the direction perpendicular to the leaf veins with a sterile scalpel and inoculated face up into various media. For the other one, seedlings with a leaf length less than 1 cm were selected, and their leaves were cut from the base with a sterile scalpel and inoculated face up into various media. The treatment groups were incubated in 1/2 MS medium with BA (0.5, 1.0 mg/L), 2,4-D (0.25, 0.5 mg/L), and TDZ (0.25, 0.5 mg/L) with KT (0.5, 1.0 mg/L) (Figure 2 and Figure 4A). The 1/2 MS medium without the addition of PGRs was set as the control group (CK). Each treatment took place in 10 glass flasks and each glass flask had 14 cross-cut leaves or 10 intact leaves. The treatment consisted of three independent replicates. After 30 days, the mortality rate (mortality rate = 100 × number of dead explants /number of inoculated explants) was calculated.

### 3.8. Plantlet Acclimatization

Glass flasks with rooted seedlings were moved to the greenhouse and cultured under natural light and temperature. After 2 days, the rooted seedlings were washed under running water to wash off the culture medium attached to the plants and transplanted to plastic pots containing mixed substrates [plant stone/ pine bark/ peat soil = 1:2:1 (*v*/*v*)]. All plants were irrigated every three days. The survival rate (survival rate = 100 × number of survival explants/number of transplanted explants) was calculated 90 days after transplanting. Each treatment consisted three independent replicates, and one replicate consisted 60 seedlings.

### 3.9. Determination of Starch and Soluble Sugar Content

Sterile explant materials at the callus stage, the PLB mass stage, the mixture of shoots and PLBs stage and the shoots stage were selected for the determination of starch content. The plant soluble sugar content assay kit (Solarbio, BC0030) and starch content assay kit (Solarbio, BC0700) used visible spectrophotometry for content determination. We weighed 0.1 g of the sample and referred to the instructions of the corresponding kit for the specific operation method to determine the content. Three biological replicates were set for each treatment.

### 3.10. Histological Analysis

We observed and photographed plant materials at different growth stages under a stereo microscope (Nikon, SMZ745T, Tokyo, Japan).

### 3.11. Data Analysis

For the statistical analysis of the data, we used IBM SPSS Statistics 20 (Microsoft Corp., Washington, DC, USA). Data were expressed as the mean ± standard error (SE) and analyzed via the one-way analysis of variance (ANOVA) followed by Duncan’s multiple range test (DMRT) at *p* = 0.05.

## 4. Discussion

### 4.1. Callus Induction and Proliferation of Paphiopedilum

The type of explants was a vital element for the success of callus induction [21]. In the present study, we failed to induce callus formations from the leaves of *P.* SCBG Huihuang90. All the leaves eventually browned and died. The results of this study were consistent with the results of other tissue cultures of *Paphiopedilum* leaves. When Long used the leaves of four species of *P. villosum* var. densissimum, *P. insigne*, *P. bellatulum*, and *P. armeniacum* as explants, various test media similarly failed to initiate the formation of calluses in the leaves, and the leaves browned and died [22]. Similarly, when Lin used leaves of a hybrid of orchids (*P. callosum* ‘Oakhil’ × *P. lawrenceanum* ‘Tradition’) as explants, viable callus induction failed [3]. So far, no successful induction of callus formation from *Paphiopedilum* leaves has been reported. Likewise, the calluses of *Paphiopedilum* are difficult to induce from other mature tissue organs. Currently, only the nodal stem of *P. rothschildianum* can be used as an explant for the induction of calluses [15].

Compared to mature tissues or organs (leaves, stem segments, etc.), the less cellularly differentiated seeds and protocorms were less difficult to dedifferentiate and more easily dedifferentiated and induced calluses. The presently regarded sources of *Paphiopedilum* callus induction are essentially seeds or protocorms [1]. Under the treatment of 2,4-D or/and TDZ, calluses have been successfully induced from the seeds or protocorms of *P. hangianum* Perner & Gruss [17], *P.delenatii*, *P. callosum* [16], *P. Alma Gavaer* [7], and a *Paphiopedilum* hybrid (*P. callosum* ‘Oakhi’ × *P. lawrenceanum* ‘Tradition’) [3], followed by differentiation to form adventitious shoots or PLBs and eventually completely regenerate plants. This was consistent with our results, that calluses of *P.* SCBG Huihuang90 can be induced from seeds rather than leaves. Callus formations were not observed in the culture of the PLB mass of *P.* SCBG Huihuang90. This was similar to the results that seeds were more efficient for callus induction than protocorms in the *Paphiopedilum* hybrid [3].

Calluses of *P.* SCBG Huihuang90 were easy to differentiate and difficult to maintain. Furthermore, 2,4-D and TDZ are plant growth regulators commonly used to induce callus formation and proliferation [1]. However, 2,4-D and TDZ failed to maintain callus status and achieve proliferation in *P.* SCBG Huihuang90, and even resulted in poor growth and significantly increased the mortality rate of calluses. Similarly, the calluses of *P. hangianum* Perner & Gruss failed to proliferate and almost all died after being treated with 1–10 mg/L 2,4-D alone [17]. In *P. Alma Gavaert*, the calluses failed to proliferate and turned pale or brown with a high dosage of 2,4-D [7]. Visibly, the damage of high concentrations of 2,4-D to *Paphiopedilum* calluses was universal. This may be because the calluses of *Paphiopedilum* are sensitive to plant hormones and their endogenous hormones are sufficient. More plant hormones upset the balance, leading to toxic effects.

By excising initial PLBs formed just after differentiation on the surface of early PLB mass, new calluses formed at the incision, achieving callus proliferation. Differentiation only restarted after the calluses had been recovered from the damaged state. Therefore, the time to complete the differentiation of calluses could be prolonged by cutting the explants, enabling the proliferation of calluses. However, the calluses proliferated limitedly after mechanical injury. So a large amount of well-grown calluses is almost impossible to obtain.

### 4.2. Propagation and Differentiation of PLBs of P. SCBG Huihuang90

The PLBs of *P.* SCBG Huihuang90 have two developmental directions. One is to proliferate to form more PLBs and the other is to differentiate to shoots with leaves. During subculture, the differentiated buds are excised, and the remaining undifferentiated PLB masses can be reused for proliferation. The shoots, differentiated from PLBs, can no longer efficiently proliferate. Therefore, it is necessary to inhibit the differentiation of PLBs.

Protocols of in vitro tissue culture through the regeneration and proliferation of PLBs have rarely been realized in *Paphiopedilum*. At present, it has only achieved success in *P. rotundifolia* [15], and BA or KT acted as a catalyst. However, PLBs of *P.* SCBG Huihuang90 tended to differentiate rather than proliferate when treated with BA or KT. The induction of PLBs of different varieties of *Paphiopedilum* have a specific demand for plant growth regulators, which makes this more challenging.

In *P.* SCBG Huihuang90, TDZ alone or in combination with 2,4-D cannot cause PLB proliferation. This might be because *P.* SCBG Huihuang90 is sensitive to exogenous plant growth regulators. TDZ is highly effective at very low concentrations [11] and so is not applicable to *P.* SCBG Huihuang90. Low concentrations of 2,4-D promote the proliferation of PLB masses. Furthermore, 2,4-D might inhibit the differentiation of PLBs, so the PLBs can maintain an undifferentiated state and have high proliferation capacity. The PLB proliferation efficiency in *P.* SCBG Huihuang90 is higher than that in *P. rotundifolia* [15].

A higher ratio of cytokinin to auxin usually promotes shoot development. However, BA and KT causes the severe browning of PLBs of *P.* SCBG Huihuang90. Cytokinins such as BA and KT have a certain effect on the accumulation of phenolic compounds [23]. The undesirable effect of cytokinins in inducing PLB differentiation exists in many orchid species. BA is neither able to induce shoots or PLB formation form alone nor in combination with NAA in hybrid *P. callosum* ‘Oakhil’ × *P. lawrenceanum* ‘Tradition’ [3]. Similarly, BA, Zea (Zeatin), and TDZ inhibited the development of PLBs in *Dendrobium huoshanense* C.Z. Tang et S.J. Cheng [24]. The effect of CW on the differentiation of adventitious buds from PLBs was better than that of plant growth regulators in *P.* SCBG Huihuang90. CW has also been found to promote the conversion of PLBs of *P. rothschildianum* into plantlets [15]. Overall, the 100% differentiation rate and high frequency of shoot formation of *P.* SCBG Huihuang90 has an advantage over *P. rothschildianum* and *P. hangianum* Perner & Gruss in terms of seedling formation [15,17]. The method described in this study is suitable for the efficient large-scale production of *P.* SCBG Huihuang90 seedlings.

### 4.3. Changes in Sugars during the Development of PLBs

Plant organ morphogenesis is a high-energy intensive process. Sugars are an important energy material for plants and constitute carbon skeletons for the biosynthesis of some cellular compounds [25,26]. The forms of sugar available in plants are diverse and their interconversion is related to the storage and consumption of energy [27]. We know nothing about the need for sugar types for the PLB regeneration process.

In *P.* SCBG Huihuang90, as the calluses differentiated and PLB masses formed, the content of starch increased significantly. Similarly, the accumulation of a certain concentration of starch is a prerequisite for the regeneration of *Humulus lupulus* var. Nugget shoot regeneration. A large amount of starch accumulates in the sites of future formations of bud primordium [28]. Also, in the early stage of the callus differentiation of *Vanilla planifolia,* starch appears in the calluses near the cell clusters that differentiate into buds or PLBs in the future [29]. The differentiation of calluses is accompanied by the increase in size and the morphogenesis of PLBs, and a large number of cells divide and differentiate. The accumulation of starch may provide energy for these physiological and biochemical behaviors.

At the early stage of the protocorm differentiation of *P.* SCBG Huihuang90, the content of soluble sugar decreased significantly, while the content of starch did not change significantly; with the development of PLBs, the content of starch and soluble sugar decreased significantly. The results indicated that the differentiation of PLBs may consume more soluble sugar, while the development of regenerated buds needs the consumption of both starch and soluble sugar at the same time. Plants with photosynthesis may no longer require excessive carbohydrate reserves after leaf morphogenesis has consumed a large amount of energy. The content of sugars such as sucrose and glucose in *V. planifolia* decreased in the early stage of shoot differentiation, indicating that sugars were consumed for the formation of vanilla shoots, which is consistent with the results of our study [29].

## Figures and Tables

**Figure 1 plants-13-00197-f001:**
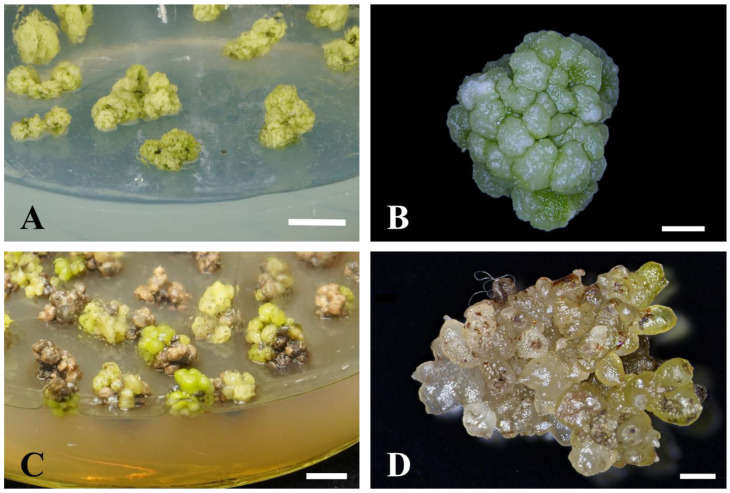
Effects of 2,4-D and TDZ on *P.* SCBG Huihuang90 PLB mass proliferation. Morphology of *P.* SCBG Huihuang90 PLB mass under 0.025 mg/L 2,4-D (**A**,**B**) and 0.75 mg/L TDZ treatment (**C**,**D**). (**A**,**C**: Scale bar = 1 cm; **B**,**D**: scale bar = 2 mm).

**Figure 2 plants-13-00197-f002:**
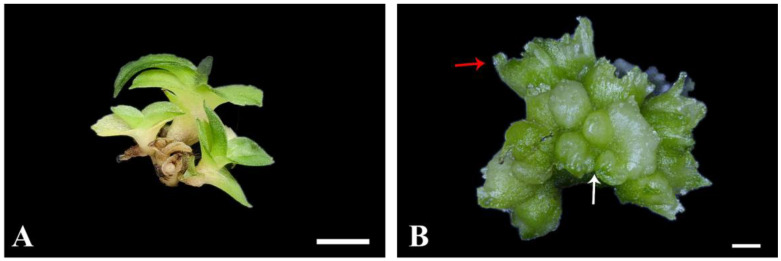
Different differentiation states of PLB mass of *P.* SCBG Huihuang90. (**A**) Morphology of cluster shoots of *P.* SCBG Huihuang90; (**B**) morphology of partly differentiated PLB mass of *P.* SCBG Huihuang90 (red arrows indicate the budding part. White arrows indicate the undifferentiated portion). (**A**: Scale bar = 50 mm; **B**: scale bar = 2 mm).

**Figure 3 plants-13-00197-f003:**
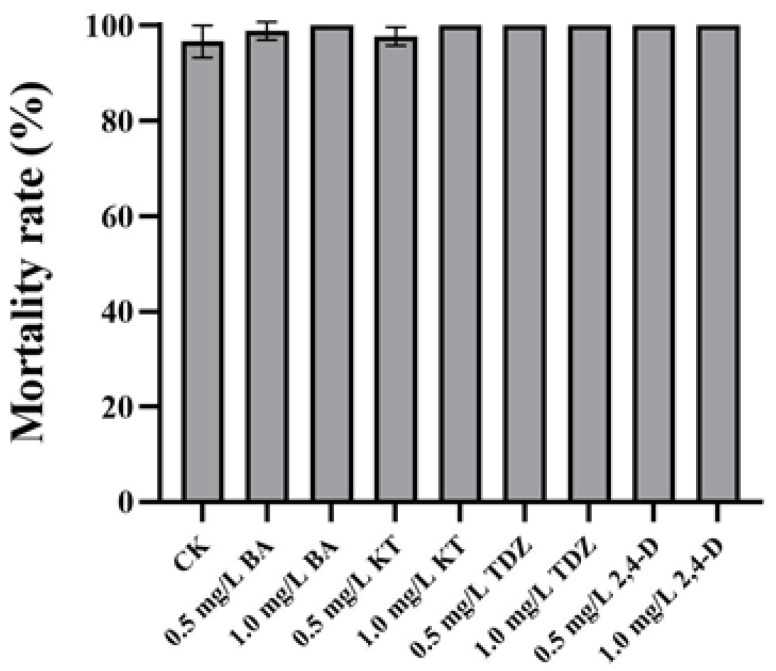
Mortality rate of cross-cut leaves of *P.* SCBG Huihuang90 at 30 days. The data represent the average of three replicates, and the error bars of each set of data show the standard error values (*n* = 3).

**Figure 5 plants-13-00197-f005:**
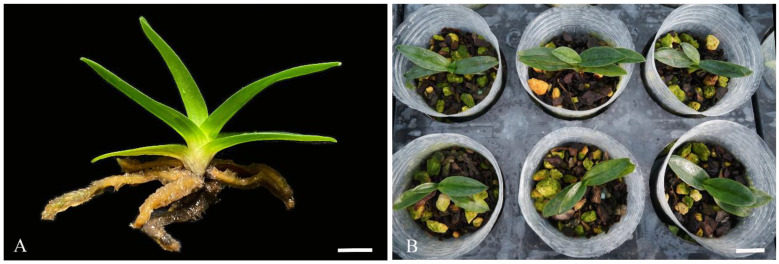
Rooting and transplanting of *P.* SCBG Huihuang90 (**A**) rooting plantlet of *P.* SCBG Huihuang90; (**B**) plantlets of *P.* SCBG Huihuang90 90 days after transplanting. (**A**: Scale bar = 1 cm; **B**: scale bar = 2 cm).

**Table 1 plants-13-00197-t001:** Effects of different concentrations of 2,4-D or TDZ on differentiation of calluses in *P.* SCBG Huihuang90.

Culture Medium	Differentiation Rate (%)	Mortality (%)
CK	82.22 ± 0.91 ^a^	17.78 ± 1.28 ^e^
0.5 mg/L 2,4-D	82.96 ± 1.39 ^a^	17.04 ± 1.96 ^e^
1.0 mg/L 2,4-D	69.63 ± 2.62 ^b^	30.37 ± 3.70 ^d^
1.5 mg/L 2,4-D	16.30 ± 3.43 ^d^	83.70 ± 4.86 ^b^
2.0 mg/L 2,4-D	0 ± 0 ^e^	100.00 ± 0.00 ^a^
0.5 mg/L TDZ	70.37 ± 1.89 ^b^	29.63 ± 2.67 ^d^
1.0 mg/L TDZ	60.00 ± 0.91 ^c^	40.00 ± 1.28 ^c^
1.5 mg/L TDZ	7.41 ± 1.39 ^e^	92.59 ± 1.96 ^a^
2.0 mg/L TDZ	0 ± 0 ^e^	100.00 ± 0.00 ^a^

Data show the mean ± standard error of three biological replicates. Duncan’s method was used for significant difference analysis between groups. Different letters in the same column represent significant differences between groups, and the same letters represent non-significant differences between groups (*p* < 0.05).

**Table 2 plants-13-00197-t002:** Effects of different concentrations of 2,4-D and TDZ on the proliferation of PLB mass in *P.* SCBG Huihuang90.

Culture Medium		Proliferation Coefficient of PLB Mass	Mortality (%)
2,4-D (mg/L)	TDZ (mg/L)		
0.00	0.00	2.66 ± 0.11 ^d^	1.85 ± 0.93 ^c^
0.01		4.66 ± 0.17 ^c^	2.78 ± 0.56 ^c^
0.025		5.76 ± 0.29 ^a^	2.78 ± 0.69 ^c^
0.05		5.04 ± 0.22 ^b^	3.47 ± 0.69 ^c^
	0.025	1.96 ± 0.49 ^d^	16.67 ± 7.22 ^b^
	0.05	2.10 ± 0.09 ^d^	13.89 ± 5.00 ^b^
	0.10	1.99 ± 0.06 ^d^	15.28 ± 2.28 ^b^
	0.25	2.82 ± 0.13 ^c^	13.89 ± 1.39 ^b^
	0.50	2.93 ± 0.08 ^c^	18.06 ± 2.78 ^b^
	0.75	2.28 ± 0.20 ^d^	30.56 ± 6.05 ^a^
0.05	0.50	1.27 ± 0.08 ^e^	27.78 ± 0.67 ^a^

Data show the mean ± standard error of three biological replicates. Duncan’s method was used for significant difference analysis between groups. Different letters in the same column represent significant differences between groups, and the same letters represent non-significant differences between groups (*p* < 0.05).

**Table 3 plants-13-00197-t003:** Differentiation of PLB mass on different media for 60 days in *P.* SCBG Huihuang90.

Culture Medium	Differentiation Rate (%)	Number of Differentiated Shoots	Browning Rate (%)
CK	33.33 ± 1.53 ^c^	8.67 ± 0.28 ^bc^	55.56 ± 2.10 ^c^
0.5 g/L AC	37.41 ± 1.54 ^c^	9.42 ± 0.22 ^b^	20.99 ± 1.63 ^e^
0.25 mg/L KT	32.10 ± 2.23 ^c^	8.26 ± 0.27 ^c^	94.44 ± 1.07 ^a^
0.5 mg/L KT	39.58 ± 3.18 ^c^	7.81 ± 0.25 ^c^	96.11 ± 0.28 ^a^
0.25 mg/L BA	39.51 ± 2.69 ^c^	8.67 ± 0.18 ^bc^	85.19 ± 1.07 ^b^
0.5 mg/L BA	59.25 ± 3.85 ^b^	8.04 ± 0.37 ^c^	88.27 ± 1.63 ^b^
10% CW (*v*/*v*) + 0.5 g/L AC	70.63 ± 2.10 ^a^	14.33 ± 0.32 ^a^	29.37 ± 2.10 ^d^

Data show the mean ± standard error of three biological replicates. Duncan’s method was used for significant difference analysis between groups. Different letters in the same column represent significant differences between groups, and the same letters represent non-significant differences between groups (*p* < 0.05).

**Table 4 plants-13-00197-t004:** Effects of different concentrations of NAA on rooting induction at 60 d in *P.* SCBG Huihuang90.

NAA (mg/L)	Percentage of Rooting Explants (%)	Average Length of the Root (cm)	Mean No. of Roots PerExplant
0	53.33 ± 0.67 ^b^	1.63 ± 0.08 ^b^	2.96 ± 0.06 ^b^
0.1	54.00 ± 3.06 ^b^	1.74 ± 0.04 ^b^	2.86 ± 0.03 ^b^
0.5	64.00 ± 2.00 ^a^	2.06 ± 0.03 ^a^	3.22 ± 0.19 ^a^
1.0	60.74 ± 0.74 ^a^	1.73 ± 0.08 ^b^	3.00 ± 0.03 ^ab^
2.0	60.67 ± 2.04 ^a^	1.80 ± 0.06 ^b^	3.14 ± 0.07 ^ab^

Data show the mean ± standard error of three biological replicates. Duncan’s method was used for significant difference analysis between groups. Different letters in the same column represent significant differences between groups, and the same letters represent non-significant differences between groups (*p* < 0.05).

**Table 5 plants-13-00197-t005:** Content of soluble sugar and starch in different growth stages of PLBs of *P.* SCBG Huihuang90.

Growth Stage	Soluble Sugar Content (mg/g)	Starch Content (mg/g)
Callus	21.95 ± 0.55 ^c^	51.15 ± 0.78 ^b^
PLB mass	50.22 ± 0.71 ^a^	69.73 ± 0.41 ^a^
Mixture of shoots and PLBs	26.02 ± 0.27 ^b^	69.74 ± 0.44 ^a^
Shoots	21.46 ± 0.27 ^c^	52.58 ± 0.54 ^b^

Data show the mean ± standard error of three biological replicates. Duncan’s method was used for significant difference analysis between groups. Different letters in the same column represent significant differences between groups, and the same letters represent non-significant differences between groups (*p* < 0.05).

## Data Availability

No new data were created.

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
