# Peer review of "Tissue Culture via Protocorm-like Bodies in an Orchids Hybrids Paphiopedilum SCBG Huihuang90"

_plants, 2024, doi:10.3390/plants13020197_

Round 1

Reviewer 1 Report

Comments and Suggestions for Authors

Review for plants-2796639

 This study established an efficient in vitro propagation protocol of a wild endangered Paphiopedilum orchid through protocorm-like bodies formed from callus derived from seed. The experimental part is well designed and executed and the results are presented clearly and discussed satisfactorily. What needs to be strengthened somewhat is the introduction, with more references to previous works on the subject.

 Abstract

L17 You say "PLBs at different developmental stages are required for different kinds of sugars" means. Probably you mean “PLBs at different developmental stages require different kinds of sugars”

Introduction

L 29-30 It is stated that: “The lack of an efficient propagation system limits the conservation and industrialization 29 of Paphiopedilum.”. Please be more specific as there is a grate number of papers reporting successful micropropagation of Paphiopedilum.

L 31 “In vitro tissue culture had the advantages”, please change “had” to “have”

L 38-39 “the callus  of Paphiopedilum were have low proliferation”, please change “were have” to “has”

L 51 please delete the word “orchids”

L 51-52 Please support the statement “However, the difficulty of PLBs  formation in Paphiopedilum during in vitro culture limits their large-scale reproduction” with appropriate references.

L 53-55 it is stated “According to reports, so far, only four species of Paphiopedilum have successfully induced  PLBs from callus and one from protocorms (Lin et al., 2000; Ng & Saleh, 2011; 54 Soonthornkalump, Nakkanong, & Meesawat, 2019; Zeng et al., 2013).” However, only on the first page of Google scholar in the search Paphiopedilum micropropagation  one can find numerous papers (eg the  following) that report microprapagation of Paphyopedilum with PLB that the authors seem to ignore.

1. Bo Long • Alex X. Niemiera • Zhi-ying Cheng •Chun-lin Long In vitro propagation of four threatened Paphiopedilum species (Orchidaceae). Plant Cell Tiss Organ Cult (2010) 101:151–162. DOI 10.1007/s11240-010-9672-1

2. Makdi Masnoddin et al., Micropropagation of an endangered Borneo Orchid, Paphiopedilum rothschildianum Callus using Temporary Immersion Bioreactor System. The Agric.Research J. Vol. 34 No. 2 (2016)

3. Some techniques in micropropagation and breeding of Paphiopedilum spp. Hoang Thanh Tung, Vu Quoc Luan, Duong Tan Nhut. DOI: https://doi.org/10.15625/2525-2518/58/4/14779

4. Tran Thai Vinh, H’ Yon Niê Bing, Dang Thi Tham, Nguyen Thi Thanh Hang, Vu Kim Cong, Nong Van Duy. Micropropagation of Paphiopedilum x Dalatense. DOI: https://doi.org/10.15625/1811-4989/14550

5. Waraporn Udomdee, Pei-Jung Wen, Shih-Wen Chin and Fure-Chyi Chen. Shoot multiplication of Paphiopedilum orchid through in vitro cutting methods. African Journal of Biotechnology Vol. 11(76), pp. 14077-14082, 20 September, 2012. DOI: 10.5897/AJB12.2047

6. In vitro propagation of Paphiopedilum orchids, S Zeng, W Huang, K Wu, J Zhang… - Critical reviews in …, 2016 - Taylor & Francis

etc…………….

L 58-60 “Few studies have been conducted to investigate the changes in the content of sugars during the development of PLBs in Paphiopedilum.” Please provide the relevant papers.

Results

L 70-72 ……the plural form of the word callus, if we were to follow strict Latin rules, would be calli, or the Anglicized form (US or UK) would be calluses. So, please use the correct form all over the manuscript.

L 117 “2.3. Effect of different mediums on PLBs differentiation”, please change “mediums” to “media” that is the plural of the Latin word medium and check this all over manuscript.

L 172 I think you mean “repetition” instead of “treatment

L 186 “affected” instead of “effected

L 191 “seedlings” instead of “seedling

L 227 “artificial” instead of “artificially

L 236 “media” instead of “medium

L 240 “culture” instead of “cultured”

L 253-254 “Each treatment with 60 explants and the experiment consisted of three independent 253 replicates.”, please rephrase

L 255 “After 60 days, observed the growth of explants and calculate the proliferation rate”, please rephrase

Please give the full name and abbreviation in parentheses the first time you mention a substance, e.g. L 259-263

L 308 “For the statistical analysis of the data used…” instead of  The statistical analysis of the data used…”

L 313 “4.1. Callus induction and proliferation of Paphiopedilum are difficult”, this cannot be a title ….

Discussion

L 314 …”profitable” it is not the right word

L 3222 “little to no induction of differentiable healing tissue formation” needs English correction

L 325 explant instead of “explants”

Comments on the Quality of English Language

English needs improvement. I have made some corrections but there are some points (I indicate them) that need to be corrected by the authors.

Author Response

Author's Reply to the Review Report (Reviewer 1)

This study established an efficient in vitro propagation protocol of a wild endangered Paphiopedilum orchid through protocorm-like bodies formed from callus derived from seed. The experimental part is well designed and executed and the results are presented clearly and discussed satisfactorily. What needs to be strengthened somewhat is the introduction, with more references to previous works on the subject.

Thank you for your suggestions and recognition. I had carefully improved the introduction as you suggested. The errors that existed throughout the text were corrected. Some higher quality images had been added or replaced. I hope that these improvements can make the quality of the article meet the requirements for publication.

Abstract

L17 You say "PLBs at different developmental stages are required for different kinds of sugars" means. Probably you mean “PLBs at different developmental stages require different kinds of sugars”

Response: It has been modified as you suggested.

Introduction

L 29-30 It is stated that: “The lack of an efficient propagation system limits the conservation and industrialization 29 of Paphiopedilum.”. Please be more specific as there is a grate number of papers reporting successful micropropagation of Paphiopedilum.

Response: The statement in the original manuscript is inappropriate, I have deleted it.

L 31 “In vitro tissue culture had the advantages”, please change “had” to “have”

Response: It has been modified as you suggested.

L 38-39 “the callus of Paphiopedilum were have low proliferation”, please change “were have” to “has”

Response: It has been modified as you suggested.

L 51 please delete the word “orchids”

Response: It has been modified as you suggested.

L 51-52 Please support the statement “However, the difficulty of PLBs formation in Paphiopedilum during in vitro culture limits their large-scale reproduction” with appropriate references.

Response: The original formulation was not rigorous, so I have made modifications (Line:52-53). The sentence that follows is also a proof of this statement. In so many species of Paphiopedilum, only six species could be successfully propagated by PLBs. PLBs was the most efficient way to propagate Paphiopedilum.

L 53-55 it is stated “According to reports, so far, only four species of Paphiopedilum have successfully induced  PLBs from callus and one from protocorms (Lin et al., 2000; Ng & Saleh, 2011; 54 Soonthornkalump, Nakkanong, & Meesawat, 2019; Zeng et al., 2013).” However, only on the first page of Google scholar in the search Paphiopedilum micropropagation  one can find numerous papers (eg the  following) that report microprapagation of Paphyopedilum with PLB that the authors seem to ignore.

  1. Bo Long • Alex X. Niemiera • Zhi-ying Cheng •Chun-lin Long In vitro propagation of four threatened Paphiopedilum species (Orchidaceae). Plant Cell Tiss Organ Cult (2010) 101:151–162. DOI 10.1007/s11240-010-9672-1
  2. Makdi Masnoddin et al., Micropropagation of an endangered Borneo Orchid, Paphiopedilum rothschildianum Callus using Temporary Immersion Bioreactor System. The Agric.Research J. Vol. 34 No. 2 (2016)
  3. Some techniques in micropropagation and breeding of Paphiopedilum spp. Hoang Thanh Tung, Vu Quoc Luan, Duong Tan Nhut. DOI: https://doi.org/10.15625/2525-2518/58/4/14779
  4. Tran Thai Vinh, H’ Yon Niê Bing, Dang Thi Tham, Nguyen Thi Thanh Hang, Vu Kim Cong, Nong Van Duy. Micropropagation of Paphiopedilum x Dalatense. DOI: https://doi.org/10.15625/1811-4989/14550
  5. Waraporn Udomdee, Pei-Jung Wen, Shih-Wen Chin and Fure-Chyi Chen. Shoot multiplication of Paphiopedilum orchid through in vitro cutting methods. African Journal of Biotechnology Vol. 11(76), pp. 14077-14082, 20 September, 2012. DOI: 10.5897/AJB12.2047
  6. In vitro propagation of Paphiopedilum orchids, S Zeng, W Huang, K Wu, J Zhang… - Critical reviews in …, 2016 - Taylor & Francis

etc…………….

Response: Thank you for the reminder. I've rechecked the articles and found two articles that include PLBs induction (including the second one you listed). I have added citations to those articles. I apologize for this mistake.

As a unique tissue structure of orchids, PLBs induced from the tissues or organs of orchids, distinguishing it from the seed-derived protocorm. The PLBs pathway has significant advantages in orchid breeding, so here we illustrate species that successfully induced PLBs. Most of the articles you listed (except the second one) were not successful in inducing the formation of PLBs. They breed Paphiopedilum by other pathways.

L 58-60 “Few studies have been conducted to investigate the changes in the content of sugars during the development of PLBs in Paphiopedilum.” Please provide the relevant papers.

Response: Incorrect wording led to a misunderstanding. No studies have been conducted to investigate the changes in the content of sugars during the development of PLBs in Paphiopedilum. We have made changes.

Results

L 70-72 ……the plural form of the word callus, if we were to follow strict Latin rules, would be calli, or the Anglicized form (US or UK) would be calluses. So, please use the correct form all over the manuscript.

Response: Response: Thank you for the reminder. I have modified them according to your suggestion.

L 117 “2.3. Effect of different mediums on PLBs differentiation”, please change “mediums” to “media” that is the plural of the Latin word medium and check this all over manuscript.

Response: Thank you for the reminder. I have modified them according to your suggestion.

L 172 I think you mean “repetition” instead of “treatment

Response: Thank you for the reminder. I have modified it according to your suggestion.

L 186 “affected” instead of “effected

Response: Thank you for the reminder. I have modified it according to your suggestion.

L 191 “seedlings” instead of “seedling

Response: Thank you for the reminder. I have modified it according to your suggestion.

L 227 “artificial” instead of “artificially

Response: Thank you for the reminder. I have modified it according to your suggestion.

L 236 “media” instead of “medium

Response: Thank you for the reminder. I have modified it according to your suggestion.

L 240 “culture” instead of “cultured”

Response: Thank you for the reminder. I have modified it according to your suggestion.

L 253-254 “Each treatment with 60 explants and the experiment consisted of three independent 253 replicates.”, please rephrase

Response: Thank you for the reminder. I have modified it and similar statements according to your suggestion.

L 255 “After 60 days, observed the growth of explants and calculate the proliferation rate”, please rephrase

Response: Thank you for the reminder. I have modified them according to your suggestion.

Please give the full name and abbreviation in parentheses the first time you mention a substance, e.g. L 259-263

Response: Thank you for the reminder. It has been modified.

L 308 “For the statistical analysis of the data used…” instead of “The statistical analysis of the data used…”

Response: Thank you for the reminder. I have modified it according to your suggestion.

L 313 “4.1. Callus induction and proliferation of Paphiopedilum are difficult”, this cannot be a title ….

Response: Thank you for the reminder. I have modified it.

Discussion

L 314 …“profitable” it is not the right word

Response: Thank you for the reminder. I have modified it.

L 322 “little to no induction of differentiable healing tissue formation” needs English correction

Response: Thank you for the reminder. I have modified it.

L 325 explant instead of “explants”

Response: Thank you for the reminder. I have modified it.

Reviewer 2 Report

Comments and Suggestions for Authors

The manuscript describes the micropropagation of an orchid. When seeds are produced, what other problems with seeds, e.g.,  %germination, plant formation, and survival etc should be mentioned? What is plantlet formation and survival of such plants? The quality of photos is poor. The results and achievements must be improved before publication. 

Major questions are:

1. explanation about seed germination and use.

2. A morphology paper must have good-quality photographs. Avoid photos of explants.

3. Analysis of carbohydrate and its justification for micropropagation?

4. Superiority of these results over previously published work.

5. Number of plants produced and survivality of plantlets etc.  

Author Response

Author's Reply to the Review Report (Reviewer 2)

The manuscript describes the micropropagation of an orchid. When seeds are produced, what other problems with seeds, e.g., %germination, plant formation, and survival etc should be mentioned?

Response: Thank you for your question. Seed germination is about 53%. We have added in the results. (Line 66)

We induced calluses directly from seeds rather than inducing them into plantlets. As described in the introduction, it is not easy to induce calluses or PLBs formation after the plantlets had formed. Therefore we did not mention plant formation. We explained this part (Line 214-216).

What is plantlet formation and survival of such plants?

Response: In the best case, one PLB can differentiate 14.33 plantlets with an induction rate of about 70.63%. We has showed this result in Table 3. The survivality of plantlets was 96% after 90 days of transplantation, which was described in "2.6 Root induction and seedlings acclimatization".

The quality of photos is poor. The results and achievements must be improved before publication.

Response: Thanks for your suggestion. Some higher quality images had been added or replaced. We have made modifications to results and achievements. I hope that these improvements can make the quality of the article meet the requirements for publication.

Major questions are:

  1. explanation about seed germination and use.

Response: Thanks for your suggestion. Seed germination is about 53%. We have added in the results (Line 66). We have added some specific instructions for seed production and use (Line 210; 214-216).

  1. A morphology paper must have good-quality photographs. Avoid photos of explants.

Response: Thanks for your suggestion. We added or changed some pictures as in Figure 1 and 2. We modified some pictures to improve the quality. Since some of the pictures are needed to illustrate the overall culture effect of the explants, I thought it would be more appropriate to use the explant pictures (e.g., Fig. 4).

  1. Analysis of carbohydrate and its justification for micropropagation?

Response: We have compared changes of carbohydrate during callus and PLB development in different orchids. We also added some explanations about justification for micropropagation (Line 393-396; 402-404).

  1. Superiority of these results over previously published work.

Response: Thanks for your suggestion. Protocols of in vitro tissue culture through regeneration and proliferation of PLBs were rarely realized in Paphiopedilum. This is the second report on the establishment of tissue culture system by PLB proliferation. We highlight this fact and add comparisons with other studies. (Line 349-350; 357-362; 374-377)

  1. Number of plants produced and survivality of plantlets etc.

Response: The number of seedlings involved in transplanting is indicated in the method (Line 281-282). The survivality of plantlets was 96% after 90 days of transplantation, which was described in "2.6 Root induction and seedlings acclimatization". Not all seeds originating from 1 capsule were used in this experiment. Therefore the number of plants produced in this experiment is not described in this manuscript. In total, we obtained about 2000 seedlings.

Round 2

Reviewer 1 Report

Comments and Suggestions for Authors

Τhe authors have made all the suggested corrections and therefore I recommend publishingaccepting the article in the present form, after two minor corrections, i.e. 

L 32 and 36: The “in vitro” should be deleted, as tissue culture is done in vitro. If it will not be deleted it should not be in italics according to MDPI instructions for authors

Reviewer 2 Report

Comments and Suggestions for Authors

The authors have incorporated all required statements about the role of seeds, germination, and offspring, survival and added new photos to justify their results and conclusion. Therefore, the manuscript is acceptable.